



# Wind Estimation based on Flight Dynamics of Unmanned Aerial Vehicle and Its Environmental Application

Dukun Chen[1,2], Weifeng Su[1], Shaojie Jiang[1,3], Honglong Yang[5], Chunsheng Zhang[5], Shutong Jiang[6], Dongyang Chang[6], Yuxin Liang[1], Hao Wang[7], Xin Yang[1], Tzung-May Fu[1], Zhenzhong Zeng[1], Lei Zhu[1], Huizhong Shen[1], Chen Wang[1], Jianhuai Ye[1,2,4,*]

[1]State Key Laboratory of Soil Pollution Prevention and Remediation for Soil Security, Southern University of Science and Technology, Shenzhen 518055, China

[2]Shenzhen Key Laboratory of Precision Measurement and Early Warning Technology for Urban Environmental Health Risks, School of Environmental Science and Engineering, Southern University of Science and Technology, Shenzhen 518055, China

[3]Jiangsu Key Laboratory of Atmospheric Environment Monitoring and Pollution Control, School of Environmental Science and Engineering, Nanjing University of Information Science and Technology, Nanjing, 210000, China

[4]Guangdong Provincial Observation and Research Station for Coastal Atmosphere and Climate of the Greater Bay Area, Shenzhen, 518055, China

[5]Shenzhen National Climate Observatory, Meteorological Bureau of Shenzhen Municipality, Shenzhen 518040, China

[6]Soarability Pte. Ltd., Singapore 409051, Singapore

[7]Shenzhen Key Laboratory for Air Vehicle and Gust Simulation, School of Mechanics and Aerospace Engineering, Southern University of Science and Technology, Shenzhen 518055, China

*Correspondence to: Jianhuai Ye (yejh@sustech.edu.cn)

Submitted to *Atmospheric Chemistry and Physics*





**Abstract.** Wind speed and direction are crucial for environmental monitoring and meteorological research, yet current measurement techniques face challenges in obtaining high spatiotemporal-resolution wind data while maintaining operational flexibility and cost-effectiveness. This study presents a wind estimation method based on attitude changes of an unmanned aerial vehicle (UAV) through controlled wind wall experiments. The estimated wind parameters were compared with measurements from an onboard wind sensor. Results from meteorological tower validations and field campaigns demonstrate that both the attitude-based and sensor-based methods achieved good agreement with reference measurements during UAV hovering. However, sensor measurements showed significant errors at high vertical flight velocities, primarily due to increased UAV downwash, while the attitude-based method maintained accuracy during flights. Building on UAV attitude changes, a machine learning algorithm was further developed to estimate wind parameters with high accuracy, offering a practical solution for future field deployments. Successful application in coastal observations showcased that wind estimation based on UAV attitude dynamics provided important spatiotemporal wind data sets that could be used to investigate the fate and dispersion of air pollutants. This work presents a reliable, sensor-free algorithm that enables low-cost, high-resolution wind measurements across diverse operational scenarios. This advancement creates new opportunities at the intersection of environmental science and emerging low-altitude economy applications, which hold promise for urban air mobility safety assessment and microscale meteorology-enhanced environmental monitoring.



**TOC**

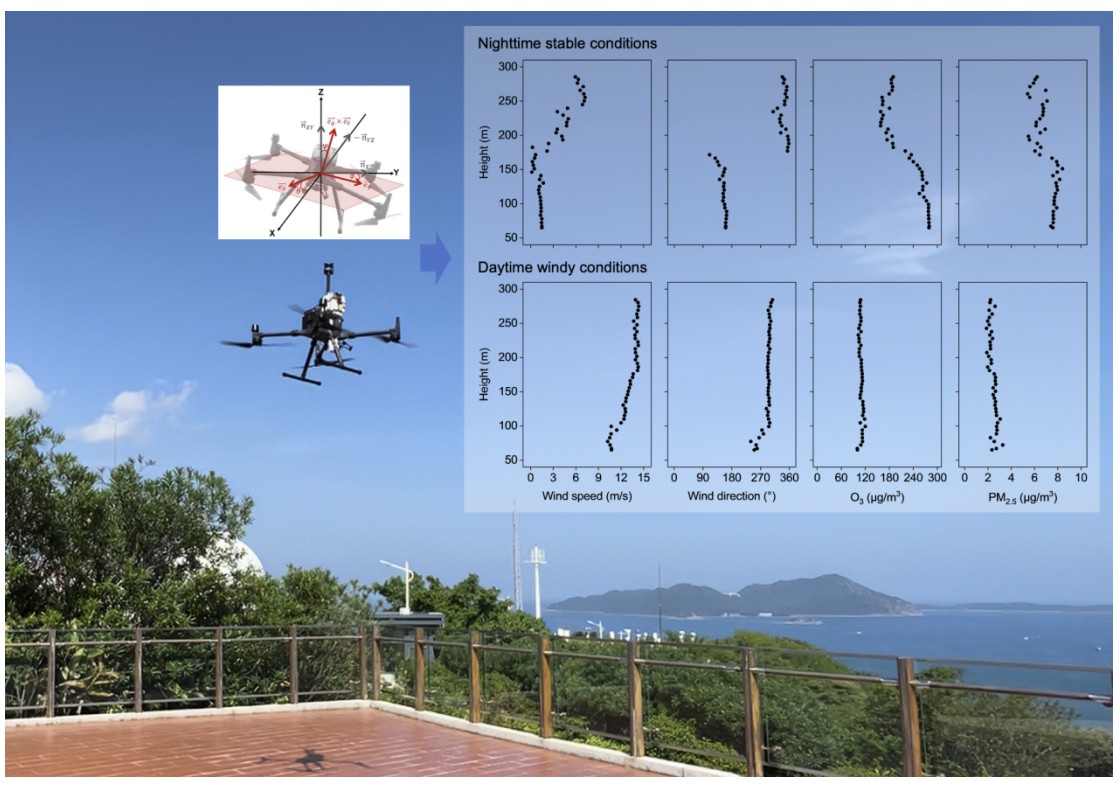



## 1. Introduction

Wind speed and direction are among the most fundamental and critical observational elements in atmospheric and environmental sciences (Yang et al., 2017; Horton et al., 2014; Wang and Chen, 2016; Guo et al., 2016; Yang et al., 2016). Accurate measurement of these parameters is vital across fields such as environmental monitoring, weather forecasting, and urban planning (Curbelo and Rypina, 2023; Yang et al., 2025a; Salmabadi et al., 2020; Alizadeh et al., 2022; Tominaga and Shirzadi, 2021). Wind direction determines the transport trajectories of air pollutants, while wind speed affects their dispersion and dilution rates. For instance, real-time wind data coupled with atmospheric dispersion models can predict smoke plume trajectories from wildfires (Curbelo and Rypina, 2023) or the spread of toxic gases from industrial emissions (Yang et al., 2025a), enabling authorities to issue timely health advisories and coordinate emergency responses. In regard to weather forecasting, wind directly influences the development and evolution of weather systems and serves as an essential parameter for numerical weather prediction models. For example, wind play a crucial role in the generation of sandstorms (Salmabadi et al., 2020; Alizadeh et al., 2022). Strong winds lift sand and dust particles from source areas, while wind direction determines the affected regions. In the context of urban planning and building design, wind patterns significantly influence thermal distribution within urban environments. Wind-driven cross-ventilation and street canyon airflow dynamics play crucial roles in regulating local microclimates and controlling air pollutant concentrations (Tominaga and Shirzadi, 2021). Strategic urban design incorporating prevailing wind direction and speed can optimize ventilation corridors and building layouts to enhance airflow, mitigate heat accumulation, and improve pollutant dispersion.

High spatiotemporal-resolution wind measurement is challenging in environmental science. Traditionally, wind speed and direction can be measured by techniques such as cup and vane anemometers, laser doppler anemometers, ultrasonic anemometers, and remote sensing (such as satellite and radar). There are significant differences among these methods in terms of accuracy, cost, and environmental suitability. For example, mechanical anemometers (e.g., cup and vane types) are widely employed in meteorological and wind energy applications due to structural simplicity and cost-effectiveness, yet their performance is constrained by factors such as low-altitude measurement constraints, dynamic response lag due to inherent mechanical inertia and physical design, turbulence sensitivity, and reduced accuracy in low air-density environments (Pindado et al., 2014; Alfonso-Corcuera et al., 2022). Optical-based systems demonstrate micro-scale resolution and multi-directional capability yet require stringent deployment conditions due to high costs and light propagation sensitivity (Lee, 2003; Diasinos et al., 2013; Knöller et al., 2024). Ultrasonic anemometers achieve high precision through non-mechanical design and wide measurement range, though they are vulnerable to temperature and humidity variations and have multipath interference (Han et al., 2008; Richiardone et al., 2012; Gaeta Lopes et al., 2017; Shan et al., 2023). Remote sensing technologies such as satellite observations provide global coverage for large-scale circulation studies but suffer from fine spatiotemporal resolution limitations (Feng et al., 2023; Hauser et al., 2023).



In recent years, unmanned aerial vehicles (UAVs) have demonstrated broad and diverse application potential in environmental science due to their unique advantages, such as low cost, flexible deployment, and the ability to obtain high-resolution three-dimensional pollution datasets (Batista et al., 2019; Zhao et al., 2021; Ye et al., 2021; Asher et al., 2021; Ye et al., 2022; Achermann et al., 2024; Li et al., 2025). UAVs can typically be categorized by wing type into rotary-wing (copter), fixed-wing, and flapping-wing configurations. Copter-type UAVs are emerging as an efficient and reliable platform for wind field measurement and monitoring (Neumann and Bartholmai, 2015; González-Rocha et al., 2023), due to their capability to hover at specific altitudes and positions, allowing for high-resolution vertical profiling at fixed locations, an advantage over fixed-wing UAVs (Li et al., 2025). This study, therefore, focuses on copter-type UAVs unless otherwise mentioned.

Current UAV-based wind estimation approaches mainly encompass three technical paradigms. The first approach involves direct measurement through onboard sensors such as anemometers. However, this method often faces challenges such as signal interference from rotor-induced turbulence and measurement inaccuracy during UAV tilt maneuvers (Liu et al., 2023; Yang et al., 2025b). The second approach utilizes mechanical model-based estimation, reconstructing wind fields through the analysis of UAV flight attitude (such as pitch and roll angles). Representative models include the dynamic particle model, kinematic particle model, and rigid body model (González-Rocha et al., 2019, 2023; Sikkel et al., 2016). While these methods provide accurate wind estimates, they are computationally expensive and usually confront inherent difficulties in precisely modeling UAV-wind field interactions under complex atmospheric conditions. The third paradigm employs data-driven analysis by established relationships between wind characteristics and UAV flight attitude (Neumann and Bartholmai, 2015; Brosy et al., 2017; Palomaki et al., 2017). Compared to the other two approaches, this method offers several advantages. For instance, it eliminates the need for additional onboard sensors, thereby reducing payload weight and lowering power consumption, simplifying system integration, and further enhancing both flight endurance and maneuverability. However, lacking a physical mechanical representation, it relies on pre-flight training data from controlled wind tests and real-world measurements (e.g., calibration against a reference instrument).

Overall, from the perspectives of system integration simplicity, cost-effectiveness, and environmental adaptability, UAV attitude-based wind estimation methods demonstrate strong potential in modern unmanned systems. While previous studies have explored these methods, critical factors affecting estimation accuracy, including payload characteristics (size and positioning), rotor-induced aerodynamic effects, and wind direction relative to UAV orientation (especially for asymmetric UAV configurations), remain understudied. These parameters significantly impact UAV attitude dynamics and subsequent wind estimation reliability. In addition, existing studies predominantly focus on wind estimation during UAV hovering or horizontal flight, paying insufficient attention to vertical wind variability. This gap is particularly significant for research on air pollutant dispersion and boundary layer dynamics. Furthermore, the lack of comparative validation under real-world meteorological conditions constrains the practical deployment of literature results. To address these challenges, this study combines laboratory wind wall experiments with field campaigns (Figures 1A-1C), aiming to systematically investigate how





the aforementioned factors affect UAV attitude-based wind estimation, as well as to explore the feasibility of using this approach for real-world vertical wind profiling. The results aim to enhance the accuracy, robustness, and operational relevance of UAV-based wind sensing for atmospheric research and environmental monitoring.

## 2. Methods

### 2.1 UAV platform

A quadcopter UAV (DJI M300 RTK) was used for the wind wall experiments and field wind measurements. The UAV has dimensions of 810 × 670 × 430 mm (L × W × H) when unfolded. The weight of the UAV is 6.3 kg with batteries. The maximum flight weight of the payload is 2.7 kg, and the maximum flight time is 55 min. The maximum ascent and descent speeds of the UAV reach 6 m/s and 5 m/s, respectively. The hovering accuracy ranges from 0.1 to 0.5 m, and the maximum tolerable wind speed is 12 m/s. The UAV is equipped with an inertial measurement unit and a GPS positioning system, which can output the attitude information required for the experiments.

### 2.2 Wind measurement and estimation

UAV-based wind speed and direction estimation was examined at Southern University of Science and Technology in Shenzhen, China, specifically within the Laboratory for Air Vehicle and Gust Simulation facility (Figure 1A). The laboratory features a specialized wind wall system capable of producing stable, controllable airflow conditions. The system can simulate wind shear and gust spectra corresponding to wind speeds up to 15 m/s in controlled environments. During the wind wall experiments, twelve wind speed levels were used by increasing the system power in 5% increments from 1 m/s to 10 m/s (specific wind speeds of 1.4 m/s, 2.2 m/s, 3.0 m/s, 3.8 m/s, 4.5 m/s, 5.3 m/s, 6.1 m/s, 6.9 m/s, 7.7 m/s, 8.5 m/s, 9.2 m/s, and 10.0 m/s). Prior to testing, the wind wall output was calibrated using a high-accuracy reference anemometer to ensure measurement reliability.

Three different UAV payload configurations were implemented to simulate real-world operational conditions, including a default setup with only the wind sensor ($M_o$), a configuration with additional payload (around 1.5 kg) on the front-top of the UAV ($M_{o+f}$), and a configuration with additional payload in the central-top ($M_{o+m}$) (Figure 1D). In order to assess the effects of wind direction relative to the UAV heading on attitude response and wind estimation, six wind directions were explored, including 0°, 45°, 90°, 180°, 225°, and 270°, as illustrated in Figure 1E. Throughout the experiments, the UAV maintained a stable hover with a flight duration exceeding 1 min for each combination of wind speed and direction. A set of relationships between wind components and UAV pitch and roll angles was obtained.





## a. Wind estimation based on UAV attitude dynamics (method 1)

UAV tilts in the presence of wind during flights. The inclination angle ($\Psi$) of the UAV, as shown in Figure 2, can be calculated using the following equations(Neumann and Bartholmai, 2015):

$$\vec{e}_\phi = \begin{pmatrix} 0 \\ \cos\phi \\ \sin\phi \end{pmatrix}, \quad \vec{e}_\theta = \begin{pmatrix} \cos\theta \\ 0 \\ -\sin\theta \end{pmatrix} \tag{1}$$

$$\Psi = \cos^{-1}\left(\frac{\vec{n}_{XY}\,(\vec{e}_\phi \times \vec{e}_\theta)}{|\vec{n}_{XY}|\,|\vec{e}_\phi \times \vec{e}_\theta|}\right) \tag{2}$$

where $\vec{e}_\phi$ and $\vec{e}_\theta$ are the vector representations of the roll angle ($\phi$, Figure 2A) and the pitch angle ($\theta$, Figure 2B), respectively; and $\vec{n}_{XY}(= (0, 0, 1))$ is the unit normal vector in the XY-plane parallel to the ground. The relationship between wind speed and UAV inclination angle can therefore be modeled using the controlled wind speed input from the wind wall system and the corresponding changes in aircraft attitude.

For wind direction estimation, the angle ($\lambda$) between the observation direction of the UAV ($-\vec{n}_{YZ} = (-1, 0, 0)$) and the projection of $\vec{e}_\phi \times \vec{e}_\theta$ on the XY-plane was first determined using Equations 3 (Figure 2C). Equation 4 resolved the position of $\vec{e}_\phi \times \vec{e}_\theta$ relative to the observation direction, and the UAV flight direction ($D_{UAV}$) was then calculated based on $\lambda$ and compass heading $\delta$ (Equation 5, Figure 2D). When hovering, the wind direction ($D_{wind}$) equals the opposite horizontal direction of UAV flight (Equation 6). When the UAV moves in the XY-plane, wind direction can be derived from the triangular relationship between wind direction, UAV flight direction, and GPS-based ground trajectory (Neumann and Bartholmai, 2015). $D_{relative}$, representing wind direction relative to UAV compass orientation, can also be obtained using Equation 7, which was set to 0°, 45°, 90°, 180°, 225°, and 270° during the wind wall experiments, as illustrated previously.

$$\lambda = \cos^{-1}\left(\frac{-\vec{n}_{YZ}\,(\vec{e}_\phi \times \vec{e}_\theta)_{XY}}{|-\vec{n}_{YZ}|\,|(\vec{e}_\phi \times \vec{e}_\theta)_{XY}|}\right) \tag{3}$$

$$P = \begin{cases} left, & \vec{n}_{XZ}\,(\vec{e}_\phi \times \vec{e}_\theta)_{XY} < 0 \\ right, & \vec{n}_{XZ}\,(\vec{e}_\phi \times \vec{e}_\theta)_{XY} > 0 \quad (P: \text{position of } \vec{e}_\phi \times \vec{e}_\theta \text{ relative to the observation direction}) \\ others, & \vec{n}_{XZ}\,(\vec{e}_\phi \times \vec{e}_\theta)_{XY} = 0 \end{cases} \tag{4}$$

$$D_{UAV} = \begin{cases} 360° - \lambda + \delta, & if\ P < 0 \\ \lambda + \delta, & otherwise \end{cases} \tag{5}$$

$$D_{wind} = D_{UAV} + 180° \tag{6}$$

$$D_{relative} = \delta - D_{wind} = \begin{cases} \lambda - 180°, & if\ P < 0 \\ 180 - \lambda, & otherwise \end{cases} \tag{7}$$





**b. Wind estimation based on onboard wind sensor (method 2)**

Wind speed and direction data were also obtained directly from a compact and lightweight ultrasonic anemometer (LI-550 TriSonica Mini, LI-COR) mounted on the UAV airframe. The anemometer determines wind parameters by measuring ultrasonic pulse transit time differences along three orthogonal axes. The accuracy of the anemometer is ± 0.2 m/s for speeds between 0 and 10 m/s, and ± 2 % for speeds from 10 m/s to 30 m/s. The wind direction measurement range is 0 to 360°, with a manufacturer-specified sensor accuracy of ± 1°. Data were recorded at 1 Hz by an onboard datalogger during the experiments.

Prior to field deployment, the sensor underwent extensive laboratory calibration in the wind wall facility to characterize its performance across the expected operational range. Calibration procedures systematically tested wind speeds from 1 to 10 m/s under three payload configurations (baseline, front-loaded, and center-loaded), while also evaluating directional response at six headings relative to the UAV (Figures 1D and 1E). This process generated detailed correction curves that accounted for rotor interference and airframe effects. The resulting calibration framework ensured reliable wind data collection during

subsequent field operations.

**c. Wind estimation using a machine learning algorithm (method 3)**

     A random forest model was developed to enable efficient wind estimation. The simulations utilized the *RandomForestRegressor* from a Python package (*sci-kit learn*). The number of decision trees was set to 100 to ensure ensemble diversity. Maximum tree depth was not restricted to capture complex data patterns. An 80:20 training-test data split and 10-

150     fold cross-validation were applied. The random seed was fixed at 42 to guarantee the reproducibility of the results.

     Model training and validation employed a dataset collected during a summer field campaign at Xichong in Shenzhen, China (Figure 1C). This is a coastal site usually selected for atmosphere-land-ocean interactions. The dataset contained measurements from 20 days (during August 21 to September 14, 2022) with 6 hovering flights conducted each day, specifically 2 flights each during morning, afternoon, and evening periods. All flights were performed over the sea surface at distances

exceeding 100 m from land. Model inputs included UAV attitude parameters (pitch, roll, and compass heading), while targets were derived from calibrated onboard ultrasonic anemometer measurements (method 2), with performance benchmarks established through comparison with attitude-based wind estimation (method 1). Performance of the random forest estimation was evaluated using the correlation coefficient ($R^2$) and root mean squared error (RMSE).

**2.3 UAV wind estimation validation**

     Validation of UAV wind estimation was conducted at the Shiyan Meteorological Gradient Observation Tower in Shenzhen, China (Figure 1B). The tower features 13 external platforms for conventional meteorological measurements. Meteorological measurement platforms are distributed across levels at 10 m, 20 m, 40 m, 50 m, 80 m, 100 m, 150 m, 160 m, 200 m, 250 m, 300 m, 320 m, and 350 m above local ground, providing multi-level wind speed and direction data. Due to



flight restrictions in the area where the maximum permissible height was 120 m, flight experiments were conducted within 100 m. Both hovering and vertical flight profiles were performed. The hovering experiment employed the default payload configuration ($M_o$) for approximately 10 min. Vertical flight experiments tested two payload configurations ($M_o$ and $M_{o+f}$) at two flight speeds (0.5 m/s and 2.0 m/s), with each configuration completing two round-trip cycles. UAV flights were conducted at least 20 m away from the tower to avoid the disturbance of UAV-induced air flows to the tower measurements.

## 3. Results and Discussion

### 3.1 UAV-based wind estimation

#### a. Estimation based on UAV attitude dynamics (method 1)

The relationship between UAV inclination angle and wind speed was modelled for different loading conditions ($M$) and
175 relative wind directions ($\eta$), as presented in Figure 3. Several fitting algorithms were evaluated, including power, logarithmic power, polynomial, and exponential functions. Among these, the power function demonstrated the best fit to the experimental data, which is expressed as:

$$V_{wind,M,\eta} = a_{M,\eta} \Psi^{b_{M,\eta}} \qquad (8)$$

where $a_{M,\eta}$ and $b_{M,\eta}$ are fitting coefficients obtained from the wind wall experiments. Overall, the fittings effectively captured
the relationship between UAV inclination angle and wind speed ($R^2 > 0.85$, Table S1).

As shown in Figure 3, the relationship between UAV inclination angle and wind speed varies significantly with relative wind direction. At identical wind speeds, inclination angles remained smaller for headwind to crosswind conditions (0°–90°) than for tailwind to rear crosswind scenarios (180°–270°). This behavior can be attributed to the advanced flight control system of the UAV used in this study. Under headwind conditions, the flight controller proactively compensates for wind disturbances
by precisely adjusting front rotor power, inducing a slight forward tilt. This active posture control leverages aerodynamic drag components to enhance stability, thereby minimizing attitude fluctuations (Ding and Wang, 2018; Otsuka et al., 2018; Lei and Lin, 2019; Jung, 2024). Conversely, during tailwind conditions, turbulent flow enveloping the airframe introduces control latency, forcing the system to apply larger attitude corrections to maintain position, ultimately amplifying the observed inclination (Ding and Wang, 2018; Otsuka et al., 2018; Lei and Lin, 2019; Jung, 2024). Adding extra payload generally reduces
how sensitive the UAV inclination is to wind, leading to smaller observed inclination angles (Figures 3B and 3C versus Figures 3A), with effects varying slightly depending on payload placement on the UAV.

The above findings differ from observations by Neumann and Bartholmai (2015), who reported that payload and wind direction had minimal effects on UAV attitude. This discrepancy likely stems from differences in UAV platforms, including





variations in design and flight control architectures. The implication is that UAV attitude-based wind estimation requires
specific algorithms tailored to the characteristics of each UAV platform.

Conventionally, without establishing the above relationships, wind speed can be estimated via force balance analysis
based on the flight attitude of the UAV (Figure 2E), as follows:

$$V_{wind} = \sqrt{\frac{2F_d}{\rho \, A_{UAV} \, C_d}} \qquad\qquad (9)$$

$$F_d = mg \, tan\Psi \qquad\qquad (10)$$

where $F_d$ is the drag force, $\rho$ is the air density, $A_{UAV}$ is the projected surface area of the UAV (perpendicular to wind direction),
$C_d$ is the drag coefficient, $m$ is the total mass of the entire UAV system, and $g$ is the gravitational acceleration.

To obtained $V_{wind}$, these equations contain two additional unknowns ($A_{UAV}$ and $C_d$) that must be resolved. While $A_{UAV}$
can typically be determined using projected area measurement tools in software such as AutoCAD, $C_d$ remains difficult to
estimate, making wind speed estimation through the force balance method challenging. Here, through the wind wall
experiments, the drag coefficient $C_d$ can be experimentally determined. An example is shown in Figure S1. During the
experiments with relative wind direction ($D_{relative}$) of 180°, the projected surface area of the UAV varied between 880 and
1120 cm² at UAV inclination within 11° over the course of the flight tests. As wind speed increased from 1 m/s to 10 m/s, the
inclination angle grew from 1° to 11°, while $C_d$ decreased from 2.75 to 0.20. For wind speed exceeded 5 m/s (corresponding
to inclination angles greater than 6°), $C_d$ stabilized at around $0.20 \pm 0.05$. These experimentally derived values can therefore
be applied in future studies for rough wind speed estimation using the force balance method for the specific UAV model (DJI
300 RTK) used in this study.

**b. Estimation based on onboard wind sensor (method 2)**

The wind sensor was also calibrated during wind wall experiments. Across varying payload configurations and wind
directions, linear relationships were consistently observed between sensor-measured wind speeds and the reference wind
speeds generated by the wind wall (Figure 4). The coefficients of these linear fits are provided in Table S2.

As shown in Figure 4, sensor measurements exhibited deviations of 30%, 15%, and 30% for the default, front payload,
and central payload configurations, respectively. The front payload configuration notably improved measurement accuracy by
reducing flight vibrations and enhancing stability (Figure 4B versus Figures 4A and 4C), consistent with the UAV flight control
system behavior described in Section 3.1a. In addition, the asymmetric sensor placement on the right front of the UAV (viewed
when facing the UAV, Figure 1E) led to maximum accuracy degradation at 225° (rear direction relative to the UAV centerline)
and secondary effects at 45° (head direction relative to the centerline).

Over the course of the experiments, the sensor registered non-zero wind speed readings even when the actual external
wind speed was 0 m/s. This phenomenon can be attributed to the rotor-induced airflow interference. During testing under





various payload configurations and wind directions, UAV rotors generated apparent wind speeds ranging from 0.1 to 1.5 m/s
(Figure 4). Similarly, the rotor interference effects on sensor measurements were most pronounced when wind approached
from the right rear (225°) and head (45°) directions relative to the UAV centerline, likely due to uneven payload distribution.

The implication of these findings is that the sensor cannot be used for field measurements without calibration, and
calibration may vary significantly with UAV model used, sensor placement, payload distribution and mass, and relative wind
direction during operation.

**3.2 Validation of UAV-based wind estimation against tower measurements**

Results of hovering flight experiments conducted at the meteorological observation tower are presented in Figure 5. Wind
speeds obtained from method 1 (based on UAV attitude) and method 2 (sensor-based) were generally consistent with each
other (Figure 5A). Both methods also closely matched the reference wind speed and direction recorded by the tower-mounted
anemometers (Figures 5B and 4C). The RMSE between the two measurement methods and the anemometer readings was less
than 0.7 m/s (for wind speed) and 20° (for wind direction), confirming the accuracy and reliability of UAV-based wind
estimation under real-world atmospheric conditions.

However, the situation changes during vertical flight operations. In this study, vertical flights were conducted at speeds
of 0.5 m/s and 2 m/s. As shown in Figure 6, wind speeds calculated using method 1 still exhibit strong agreement with
meteorological tower measurements, regardless of whether the UAV ascends or descends at 0.5 m/s (Figure 6A-I) or 2.0 m/s
(Figure 6B-I). For the sensor-based method (method 2), the measurements matched the tower data at the lower vertical speed
of 0.5 m/s (Figure 6A-II). However, when the vertical speed increased to 2.0 m/s, significant deviations became apparent
(Figure 6B-II). These discrepancies are likely caused by increased rotor-induced turbulence during high-speed vertical flight,
which degrades sensor measurement accuracy. Notably, the addition of payload had no effect on the wind estimation accuracy
when using method 1 (Figures 6B-I and 6D-I), while it amplified the impact of rotor-induced airflow disturbances on sensor
measurements (Figures 6B-II and 6D-II). Similar findings were observed for wind direction estimations (Figure S2).

These comprehensive analyses demonstrate that the attitude-derived method (method 1) robustly estimates wind variables.
For the sensor-based method (method 2), results indicate that with proper calibration, sensors can achieve accuracy comparable
to commercial meteorological instruments during UAV hovering operations. However, the sensor-based method shows
limitations for vertical profiling applications due to its significant susceptibility to rotor-induced airflow disturbances. These
systematic errors were consistently observed across all test configurations. Nevertheless, the successful validation of the
attitude-based method substantially enhances the potential for UAV applications in atmospheric research, offering distinct
advantages for measurements in locations inaccessible to conventional tower-based systems and in scenarios requiring rapid
deployment of mobile platforms.



**3.3 Field wind measurements**

Field flight experiments were further conducted at a coastal site in Shenzhen, China, to evaluate the performance of methods 1 and 2 for wind estimation. For analytical simplicity, only UAV hovering data was utilized in this comparative analysis.

As shown in Figures 7A and S3, strong agreement was observed between the two methods, with median differences of approximately 0.1 m/s for wind speed and less than 10° for wind direction, confirming the reliability of the UAV attitude-based approach. To enhance the wind estimation methodology, we further implemented a machine learning framework (method 3) using UAV attitude parameters as inputs and the corrected wind sensor measurements as training outputs. The model achieved excellent predictive performance with $R^2$ values exceeding 0.90 for both training and test datasets (Figures 7B and Figure S4). When applied to independent datasets from the meteorological tower flight experiments (Section 3.3), the estimates maintained good agreement with tower measurements (Figure S5). This consistency across different validation approaches confirms the robustness of the UAV attitude-based methodology and its potential for practical applications in wind measurement.

Wind profile estimation offers critical insights into atmospheric pollutant transport and dispersion dynamics. Using the field campaign data as an example, we constructed diurnal vertical wind profiles by analyzing UAV attitude variations during flight operations. During the field campaign, we also collected vertical pollutant concentration profiles using a pre-calibrated sensor package (Sniffer V2, Soarability Pte. Ltd.), with cross-validation against reference instruments at a ground station located 50 m from the flight site.

Representative vertical profiles of both wind and pollutant measurements are displayed in Figure 8, which reveals the crucial influence of wind profiles on pollutant distribution patterns. For instance, persistently low wind speeds were observed throughout the day of August 23, indicating stable atmospheric conditions which were conducive to pollution accumulation. Pollutant concentrations in this day exhibited typical diurnal variations, gradually increasing from morning, peaking in the afternoon due to photochemical activity, and decreasing at night with reduced emissions and photochemical processes. In contrast, September 7 featured strong winds that enhanced pollutant dispersion, resulting in consistently low pollution levels.

The September 14 case demonstrated a complex vertical wind structure, with speeds decreasing from morning to nighttime minima while increasing with height. Pollutant concentrations varied significantly with wind direction changes. Notably, at 19:00, a 180° wind shift transported polluted air masses from the south, sharply increasing observed concentrations. Surface cooling and calm winds at night created stable stratification, trapping pollutants near the surface and producing distinct vertical gradients. The nocturnal boundary layer height (150-200 m), identifiable from wind and pollutant profiles (Guimarães et al., 2019, 2020; Ye et al., 2021), showed reduced pollutant concentrations at the residual layer due to enhanced wind speed and dispersion. A subsequent wind shift to 360° brought back cleaner northern air, reducing both surface concentrations and

 

vertical gradients. These observations underscore the importance of vertical wind profiling enabled by UAV-attitude-based estimation, for understanding atmospheric transport mechanisms and pollution dynamics.

**4. Atmospheric Implications**

This study develops a UAV wind estimation method based on attitude changes. Validation through wind wall and field experiments demonstrated the reliable performance of the attitude-based approach. Key findings indicate that payload variations significantly affect attitude responses, with distinct patterns observed across different wind directions, underscoring the importance of comprehensive training data to improve model accuracy. We further developed a supervised learning framework to extract wind parameters directly from UAV attitude data. The machine learning model achieved accurate
predictions of both wind speed and direction while maintaining practicality for field deployment. Compared to the attitude-based approach, the results revealed significant rotor-induced interference when using onboard sensors for wind measurement, particularly during vertical maneuvers. This highlights the need for pre-deployment calibration and bias corrections for sensor measurements. Collectively, these results demonstrate the strong potential for precise, sensor-free wind field estimation using UAV attitude data.

The UAV attitude-based wind estimation method, while promising, presents several limitations that require future consideration. First, this approach requires establishing accurate relationships between UAV inclination angles and wind speed, which may vary across different UAV models. For example, the DJI 300 RTK used in this study exhibited substantial payload and relative wind direction effects, a phenomenon not observed by Neumann and Bartholmai (2015) with a different UAV platform. Thus, UAV-specific relationships must be developed before field deployment for future studies. Additionally, this
study identified greater uncertainty in the attitude-based wind speed estimates below 2 m/s under headwind conditions (e.g., 45° to 90°, Figure 3, likely owing to the advanced UAV flight control system described in Section 3.1). To address this challenge, future research could explore hybrid approaches integrating attitude-derived estimates with measurements from pre-calibrated onboard sensors, thereby enhancing accuracy in low-wind-speed regimes. Finally, creating a comprehensive database linking UAV attitude data to wind measurements across diverse flight conditions (e.g., hovering, horizontal, and
vertical flight at varying speeds) would be highly valuable. Such a dataset would enable the training of advanced AI models, accelerating the development of reliable, attitude-based wind field prediction methods. Although literature has reported using machine learning to train wind observation data, these efforts have typically been constrained by the availability of single flight tests or very limited datasets (Zhu et al., 2025), which fail to capture the full variability of UAV responses under diverse operating conditions. In practice, UAV attitude and rotor dynamics are strongly modulated by wind direction, flight mode, and
payload configuration, leading to highly nonlinear and platform-specific responses, as demonstrated in this study. Expanding datasets and integrating physical knowledge of UAV aerodynamics into data-driven models will therefore be essential for building more robust and transferable AI-based wind sensing frameworks.



With the global development of the low-altitude economy (Huang et al., 2024; Saadé et al., 2025; Tan et al., 2025; Zhou, 2025), UAV attitude-based wind estimation has become an essential enabling technology. The sensor-free approach presented
in this study utilizes inherent flight dynamics to generate reliable wind field data, providing advantages for low-altitude operations where conventional measurement methods encounter limitations. Due to its operational simplicity and cost-effectiveness, this technique proves particularly valuable for widespread implementation across crucial low-altitude economic sectors such as urban air mobility systems for flight safety assurance. For example, the method presented in this study integrates easily with existing UAV operations, positioning it as a fundamental innovation for low-altitude economic activities that rely
on real-time three-dimensional wind field data.

In environmental applications, UAV platforms offer distinct advantages by providing precise, flexible, and efficient wind field measurements with high spatiotemporal resolution. This technological progress transforms environmental monitoring approaches by enabling vertical wind profiling, which significantly improves the analysis of atmospheric pollutant transport patterns. These enhancements allow more accurate pollution source identification and support the development of targeted
mitigation strategies, particularly in urban environments where building configurations and street-level airflow interactions critically influence local air quality. The three-dimensional wind data obtained from UAV measurements can inform urban planning decisions by characterizing how architectural geometries modulate near-surface ventilation efficiency. Furthermore, the measurement capabilities provide critical data for validating high-resolution weather and climate models, especially for simulating complex urban canopy effects on microscale wind circulation patterns that govern heat dissipation.

Beyond urban research, UAV-based wind estimation opens new opportunities for field studies in natural ecosystems and remote regions. For example, reliable vertical wind profiles can substantially improve the quantification of forest canopy–atmosphere exchange processes, including the dispersion of biogenic volatile organic compounds and greenhouse gases (Jiang et al., 2024; Ye et al., 2021). Similarly, accurate wind field characterization over coastal and marine environments enhances the interpretation of air–sea exchange fluxes, sea-breeze circulation, and the long-range transport of marine aerosols (Zhao et
al., 2021). For atmospheric chemistry studies, UAV-derived wind fields provide essential inputs for constraining dispersion models and for interpreting aircraft or ground-based observations, particularly in regions where conventional meteorological measurements are sparse or absent (Ye et al., 2022).

Through these varied applications, UAV-based wind measurement technology is emerging as an innovative tool that connects the low-altitude economy with environmental science. By delivering spatiotemporally resolved wind data in complex
terrains and under diverse atmospheric conditions, the approach not only advances intelligent environmental risk management but also supports sustainable development initiatives and climate adaptation strategies on regional to global scales.





**Supplemental Materials.** The supplemental materials include the following items:

Table S1    Fitting coefficients for UAV inclination-wind speed relationships.

Table S2    Fitting coefficients for wind sensor calibrations.

Figure S1    Relationship between input wind speed and UAV parameters.

Figure S2    Comparison between UAV-based wind direction estimates and reference measurements from the meteorological observation tower.

Figure S3    UAV-based wind direction estimation for the field observation campaign.

Figure S4    Performance of wind estimation using machine learning algorithms.

Figure S5    Comparison between machine-learning-based wind speed estimates and reference measurements from the meteorological observation tower.

**Corresponding Author.** Jianhuai Ye (yejh@sustech.edu.cn), ORCID: 0000-0002-9063-3260

**Author Contribution.** J.Y. designed the research; D.C., W.S., S.J., Y.L. and J.Y. conducted wind wall and meteorological tower flight experiments; S.J. and J.Y. performed the UAV field campaign; D.C. and J.Y. analyzed the data and wrote the original manuscript; all authors contributed to the data interpretation and manuscript writing.

**Competing Interest.** The authors declare no competing financial interest.

**Acknowledgments.** This work was funded by National Key Research and Development Program of China (2024YFC3714300), National Natural Science Foundation of China (Nos. 42375091, 42105098, and U24A20515), and Shenzhen Science and Technology Program (JCYJ20241202152804007 and KQTD20240729102048052). High Level of Special Funds (G03050K001) from Southern University of Science and Technology is acknowledged.



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



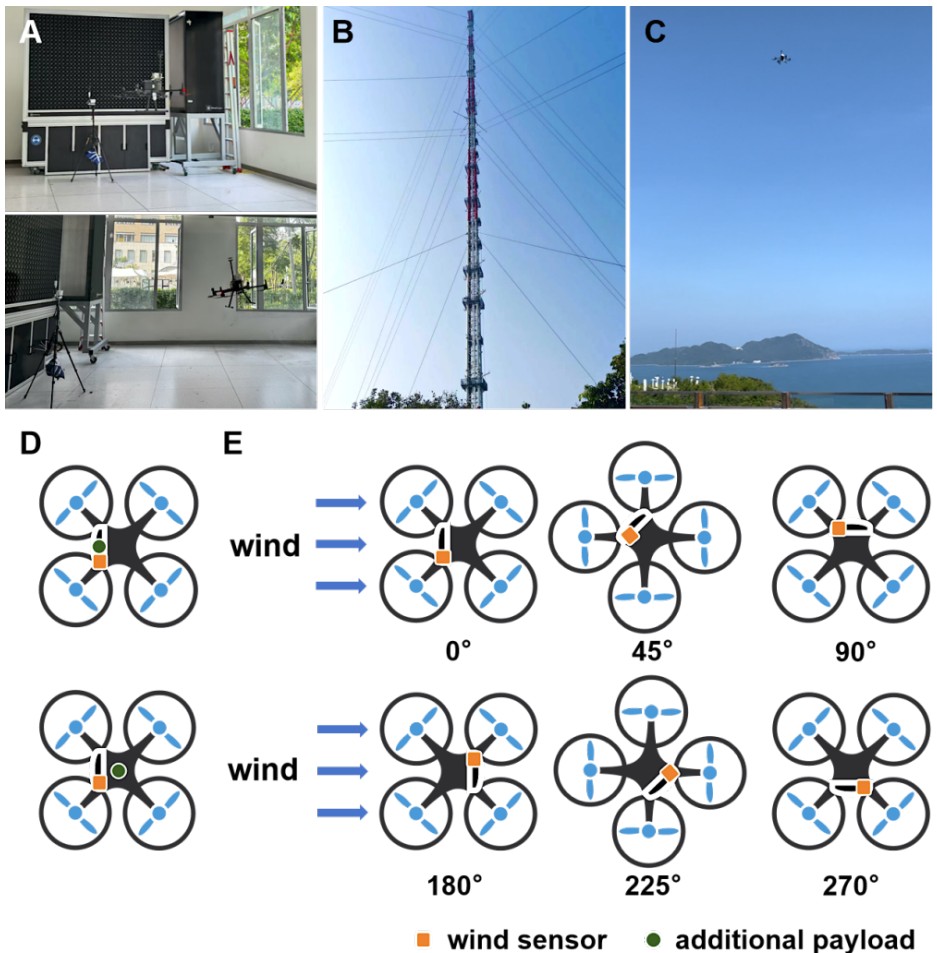

**Figure 1**   UAV flights conducted in a wind wall laboratory (A), at a meteorological observation tower (B), and at a coastal site (C). Schematic diagrams of UAV payload configuration (D) and UAV flights under different relative wind directions.



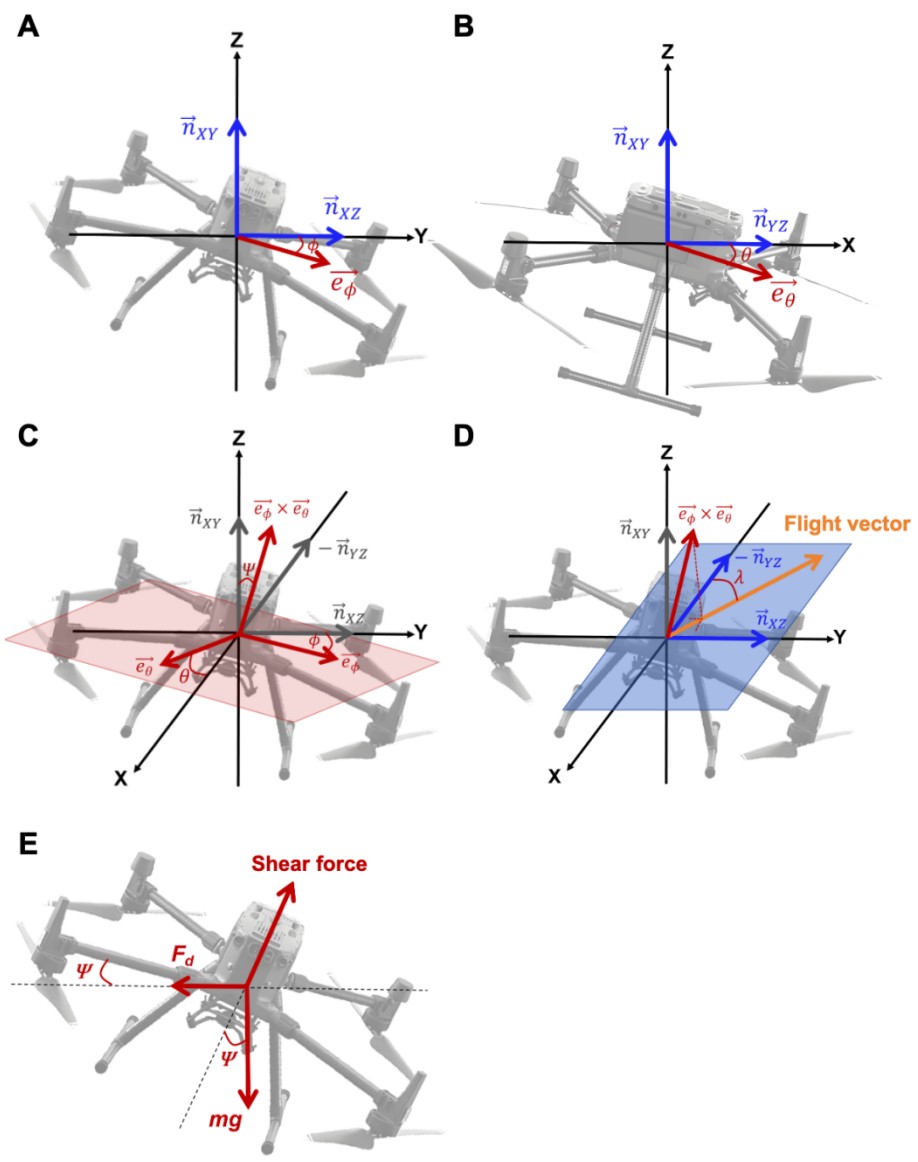

**Figure 2**    Schematic diagrams of UAV attitude coordinates (A-D) and force balance analysis (E).



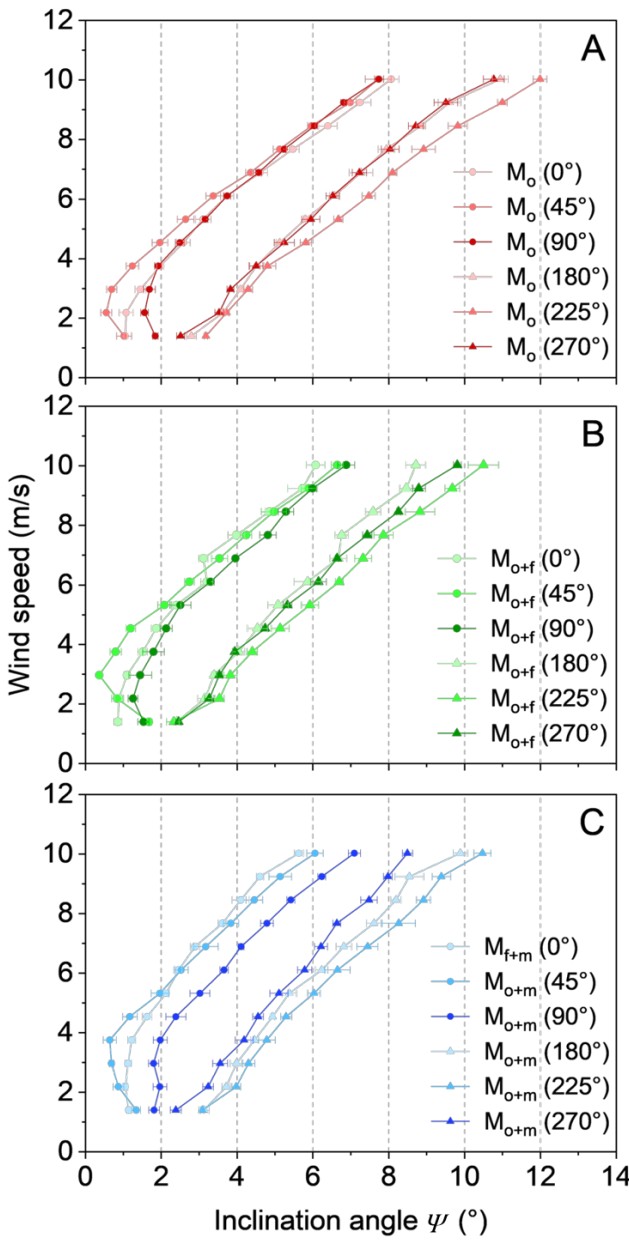

**Figure 3** Relationship between UAV inclination angle and wind speed under varying payload (A: default, B: additional front-top payload, C: additional central-top payload) and wind direction (0°, 45°, 90°, 180°, 225°, and 270°) conditions. The payload configurations and relative wind directions are illustrated in Figure 2. The relationships are characterized by power function fits, with coefficients for each flight scenario provided in Table S1.





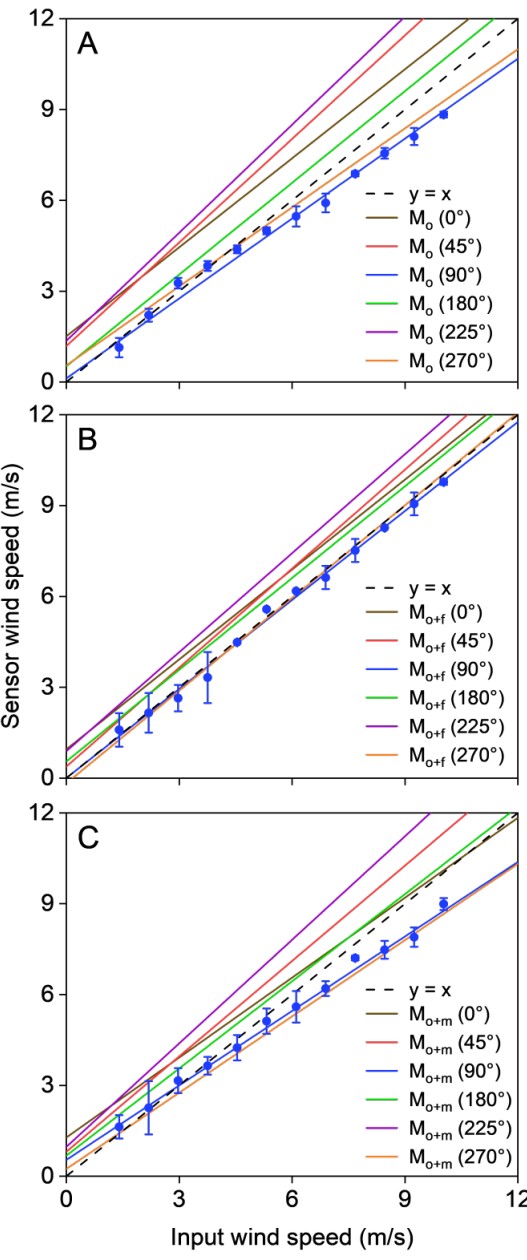

**Figure 4** Relationship between input wind speed and sensor-measured wind speed under varying payload (A: default, B: additional front-top payload, C: additional central-top payload) and wind direction (0°, 45°, 90°, 180°, 225°, and 270°) conditions. Both measurements and fitted curves are shown for 90° relative wind direction scenarios, while only fitted curves are presented for other directions. Corresponding fitting parameters are provided in Table S2.





**Figure 5** Comparison between UAV-based wind speed estimates (from methods 1 and 2) and reference measurements from the meteorological observation tower during hovering flight operations.





**Figure 6**  Comparison between UAV-based wind speed estimates and reference measurements from the meteorological observation tower during vertical flight operations: ascending and descending at 0.5 m/s (A) and 2 m/s (B) with default payload, and at 0.5 m/s (C) and 2 m/s (D) with additional front-top payload. Gray shaded areas indicate hovering periods.



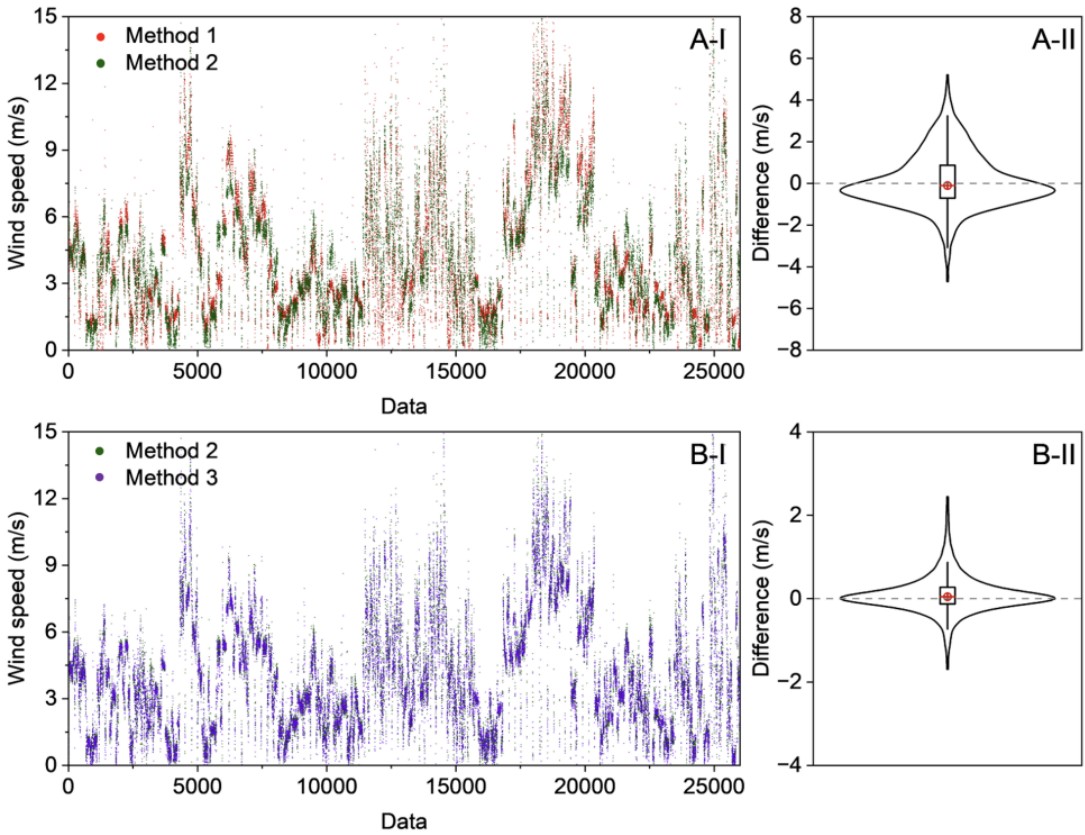

**Figure 7** UAV-based wind speed estimation and deviation analysis comparing methods 1 versus 2 (A-I, A-II) and methods 2 versus 3 (B-I, B-II) from the field observation campaign.





**Figure 8**    2-D contour plots of the vertical profiles of wind speed, wind direction, O₃ and CO concentrations measured on August 23, September 7, and September 14 of 2022, respectively.