# Peer review of "Wind Estimation based on Flight Dynamics of Unmanned Aerial Vehicle and Its Environmental Application"

_EGUsphere, 2025_

## Referee Comment (RC2)

**Referee Report on**

**Wind Estimation based on Flight Dynamics of Unmanned Aerial Vehicle and Its Environmental Application**

The article presents a wind estimation approach that uses the pitch and roll of a rotorcraft UAV to estimate the wind speed and direction. The authors present three methods utilizing the aircraft kinematics based approach of Neumann and Bartholmai (2015), a direct approach using an aircraft-mounted sensor, and a machine-learning-based approach using the pitch, roll and compass heading. The authors find good agreement of all three approaches during hovering, whereas the aircraft-mounted sensor was inaccurate during ascent/descent.

My primary concern with this paper is that it does not present much in the way of new findings or information. The kinematics-based approach appears to be very similar, if not identical, to the approach developed by Neumann and Bartholmai (2015); it is very common practice to deploy sonic anemometers on rotorcraft UAVs; and calibrations using wind tunnels/wind walls against towers are also standard practice. Furthermore, the authors strongly imply that the approaches that they use are only suitable for this specific aircraft configuration (lines 193-195) which limits the applicability of much of the presented results to this specific aircraft. The new material here therefore appears to be the experimental results presented in Figure 8 and the machine-learning-based approach for wind estimation, which is only very sparsely presented and not thoroughly validated.

I suggest that the authors specifically identify how their kinematics-based approach differs from that of Neumann and Bartholmai (2015), or alter the title of their manuscript to better reflect that this a test of existing technology, rather than a new development.

Additional clarifications should also be provided within the manuscript. Specifically:

- 1. The authors would do well to include the paper:
  - Ahmed, Zakia, Mekonen H. Halefom, and Craig Woolsey. "Tutorial review of indirect wind estimation methods using small uncrewed air vehicles." Journal of Aerospace Information Systems 21.8 (2024): 667-683."
  - in their literature review as it provides a relatively recent summary of existing wind estimation approaches.
- 2. Much more detail is required regarding the anemometer placement. The sketch provided in Figure 1 is insufficient to assess the location of the anemometer relative to the rotor plane, which is crucial for ensuring it is free of inflow effects. Given that the authors clearly noticed inflow effects, It appears that the anemometer may have been placed to near the rotor plane, which has long been acknowledged as introducing significant inaccuracies in the wind measurement. However, the reader is unable to assess the suitability of the authors' sensor placement with the information provided in the manuscript.
- 3. What are the weight and dimensions of the 'additional payload'? This information is required to ensure repeatability of the results.
- 4. Lines 118-120: "The relationship between wind speed and UAV inclination angle can therefore be modeled using the controlled wind speed input from the wind wall system..." This statement is an assumption and is not directly supported by equations 1 and 2 (velocity does not appear implicitly in these equations)
- 5. Line 122: Did the authors intend to refer to figure 2D instead of 2C?  $\psi$  only appears in figure 2D.
- 6. Lines 197-212: Equation 9 neglects the component of thrust directed by the rotors in the horizontal plane (i.e. the reason the aircraft produces  $\psi$  during station-keeping/hovering). The remainder this then casts doubt on the remainder of the discussion of the drag-force-based wind estimation and I strongly suggest the authors remove this section of text, particularly since this approach was not applied or used anywhere else in the manuscript.
- 7. The aircraft's moment of inertia will play a large role in the time response to wind gusts. Do the authors have any sense as to what the response times are for their aircraft configuration?

- 8. In figure 6, the authors ascend and descend, for what looks like 100 m (presumably ascending from the ground to the 100 m measurement height and back). However they state in the text that they only used the 100 m measurement position for validation. Is that correct? The vertical wind gradients are not negligible. The authors need to provide more details about this test in order for the reader to understand it further. Particularly since ascent/descent appears to be their primary mode of application in Fig. 8 and they are inferring a lot from this validation test.
- 9. Lines 265-267 tout the consistency of the three methods tested in Figure 7 a indicating successful wind measurements. However, figure 7 Shows that there are many points which lie outside the general trend (most noticeable between 0-3 m/s between 17500 and 20000. These are likely incorrect estimates. Why do these outliers appear on all three methods? The fact that these outliers also appear in the machine-learning-based approach likely reflects the use of these measurements as training data. Therefore, this consistency may simply be an artifact of the interwoven calibration approaches used for the different techniques.